

# Survival predictors of lung cancer patients in ICU: the importance of acute kidney injury prediction and prevention

Jue Shen[1], Changsong Wang[2], Gang Ma[3], Hong-Zhi Wang[4], Xuezhong Xing[5], Biao Zhu[6], Jianghong Zhao[7], Donghao Wang[8] and Mingou Cui[1]

[1] Department of Intensive Care Unit, Zhejiang Cancer Hospital, Hangzhou Institute of Medicine (HIM), Chinese Academy of Sciences, Hangzhou, China
[2] Department of Intensive Care Unit, Harbin Medical University Cancer Hospital, Harbin, China
[3] Department of Intensive Care Unit, Sun Yat-sen University Cancer Center, Guangzhou, China
[4] Department of Intensive Care Unit, Beijing Cancer Hospital, Beijing, China
[5] Department of Intensive Care Unit, Cancer Hospital Chinese Academy of Medical Sciences, Beijing, China
[6] Department of Intensive Care Unit, Fudan University affiliated Shanghai Cancer Hospital, Shanghai, China
[7] Department of Intensive Care Unit, Hunan Cancer Hospital, Changsha, China
[8] Department of Intensive Care Unit, Tianjin Medical University Cancer Institute and Hospital, Tianjin, China

Corresponding authors
Donghao Wang,
wangdonghao@tjmuch.com
Mingou Cui, cmoxlj@163.com

## ABSTRACT

**Background**. To identify potential predictors of short-term survival among patients with lung cancer admitted to the intensive care unit (ICU).

**Methods**. A multicenter longitudinal observational study of patients with lung cancer was conducted between May 10, 2021 and July 10, 2021, at the ICUs of 37 cancer-specialty hospitals in China. This study included patients with a primary diagnosis of lung cancer who were admitted to the ICU for $\geq$24 h. Predictive factors for ICU outcomes, with 90-day survival as the major outcome, were explored using single and multivariate analyses.

**Results**. A total of 269 patients were included in the final analysis. The 90-day mortality rate following ICU care was 45.4%. Patients with 90-day mortality exhibited more severe conditions before admission, a higher number of ICU-related complications, and underwent more intense treatment than survivors. Notably, despite the low recorded incidence, acute kidney injury (AKI) was independently associated with ICU, in-hospital, and 90-day mortality outcomes in the multivariate analysis. Furthermore, condition severity at admission and ICU treatment choices, especially anti-infection regimen, were identified as potential correlators of a higher AKI risk.

**Conclusion**. AKI prediction and prevention may require prioritization in patients with lung cancer admitted in the ICU.

## INTRODUCTION

Patients with cancer may require intensive care unit (ICU) admission owing to complications or treatment-associated side effects (*Koutsoukou, 2017*; *Martos-Benítez et al., 2020*). The progression of cancer treatment and ICU management has significantly improved the survival outcomes of patients with cancer. Conversely, longer disease duration

and treatment-related events have also increased the number of patients subjected to ICU care (*Azoulay et al., 2017*; *Puxty et al., 2015*). Lung cancer represents the most common cause of cancer-related mortality. Additionally, patients with lung cancer may exhibit higher risks of acute respiratory failure and bacterial infection (*Cupp et al., 2018*; *Williams, Ford & Coopersmith, 2023*). Therefore, lung cancer remains the most common cancer type among ICU-admitted patients, with solid tumors exhibiting the poorest prognosis (*Zarogoulidis et al., 2013*).

Previous studies have reported that the short-term mortality rate of ICU-admitted patients with solid tumors averaged approximately 20%–30% (*Martos-Benítez, Soto-García & Gutiérrez-Noyola, 2018*; *Ostermann et al., 2017*; *Puxty et al., 2014*). Studies in ICU-admitted patients with lung cancer have reported various results. A study in newly diagnosed patients with lung cancer who underwent ICU treatment revealed an overall mortality rate of approximately 18.7% (*Park et al., 2021*). Another study reported ICU and hospital mortality rates of 36% and 51%, respectively, among patients with lung cancer (*Soubani & Ruckdeschel, 2011*). An analysis of Medical Information Mart for Intensive Care III, a large-scale single-center database, revealed that ICU-admitted patients with lung cancer had 28-day in-hospital and 6-month mortality rates of 30.6% and 68.2%, respectively (*Qian et al., 2023*). For ICU-admitted patients with advanced or metastatic lung cancer who required mechanical ventilation, the 3-month mortality rate was as high as 67% (*Barth et al., 2018*).

Owing to the high risk of poor outcomes among ICU-admitted patients with lung cancer, identifying those with more probability to benefit from ICU admission and the timely recognition of prognostic signs during ICU management is crucial. Determining the predictors of these patients may help optimize the admission and treatment strategies. The prognostic characteristics of ICU-admitted patients with lung cancer remain not well elucidated. General factors that are correlated with worse outcomes encompass advanced oncologic disease, poor performance status, and signs of critical complications (*Cuenca et al., 2022*; *Shen et al., 2022*). Indicators noted in various studies lack consistency and are very limited in clinical practice (*Adam & Soubani, 2008*; *Jennens et al., 2002*; *Soares et al., 2004*; *Zarogoulidis et al., 2013*).

Acute kidney injury (AKI) is among the common complications of patients with cancer and has a profound influence on prognosis (*Gupta, Gudsoorkar & Jhaveri, 2022*). AKI has been suggested to be a significant prognostic factor for ICU patients (*Alba Schmidt et al., 2024*; *Hashemian et al., 2016*). Moreover, AKI has been reported to be a significant risk factor for the short-term mortality of ICU-admitted patients with cancer (*Nazzal et al., 2022*; *Seylanova et al., 2020*; *Yuan et al., 2020*). However, studies on AKI incidence and influence on ICU-admitted patients with lung cancer are very limited.

Using the data from a large-scale multicenter cross-sectional study of ICU-admitted patients with cancer, we analyzed the potential predictors of the short-term survival of patients admitted to the ICU. AKI was identified as a critical predictor for the short-term mortality of these patients.

## MATERIALS AND METHODS

### Study design and population

From May 10, 2021, to July 10, 2021, a multicenter cross-sectional study was conducted at the ICUs of 37 cancer-specialty hospitals in China to gather data on the characteristics of patients with critical illness. The original study included patients with various cancer diagnoses who aged ≥14 years and were admitted in the ICU for ≥24 h. In the current study, the clinical records of patients with primary lung cancer diagnosis in the original database were reviewed. To explore the predictors of short-term mortality, characteristics at baseline and during ICU management were analyzed. The ethics committee of the Tianjin Medical University Cancer Institute and Hospital approved this study (bc2021065), and the procedures were performed following the ethical standards of the responsible committee on human experimentation and with the Helsinki Declaration of 1975. Written consents was obtained from the patients.

### Data collection

Data extracted from the clinical records encompassed general demographic information (age, sex, and body mass index (BMI)), clinical history (source of admission, primary diagnosis, and cancer treatment history), Sequential Organ Failure Assessment and Acute Physiology and Chronic Health Evaluation II scores at admission, ICU critical condition diagnosis and treatment applied, ICU and in-hospital outcomes (delirium, ICU duration, ICU death, and in-hospital death), and 90-days survival follow-up.

AKI diagnosis and grading were made according to the 2012 KDIGO criteria (*Kellum & Lameire, 2013*). Briefly, AKI was diagnosed when any of the following laboratory test results were identified: an increase in serum creatinine (SCr) levels by ≥0.3 mg/dL (≥26.5 µmol/L) within 48 h; an increase in SCr levels to ≥1.5-fold from the baseline, which is known or presumed to have occurred within the prior 7 days; or a urine volume <0.5 mL/kg/h for 6 h. Grade I was defined as SCr levels of 1.5–1.9-fold higher than the baseline or an increase of ≥0.3 mg/dL (≥26.5 µmol/L), with a urine output of <0.5 mL/kg/h for 6–12 h. Grade II was defined as an SCr level of 2.0–2.9-fold higher than the baseline, with a urine output of <0.5 mL/kg/h for ≥12 h. Grade III was graded when the SCr level was 3.0-fold higher than the baseline or an increase in the SCr levels to ≥4.0 mg/dL (≥353.6 µmol/L) or the initiation of renal replacement therapy or in patients <18 years a decrease in eGFR to <35 mL/min per 1.73 m$^2$, with a urine output of <0.3 mL/kg/h for ≥24 h or anuria for ≥12 h.

### Statistical analysis

To evaluate the normality of the quantitative data distribution, the Shapiro–Wilk test was performed. Owing to the skewed distribution, the median and quartile ranges (P25–P75) were used to describe the quantitative data. Categorical data were described using frequency (*n*) and percentage (%). Quantitative data were compared among the groups using the Kruskal–Wallis H test. Categorical data were compared between the groups using either the $\chi^2$ test or Fisher exact test. To analyze the correlated factors of ICU, in-hospital, and 90-day mortality outcomes, single and multivariate logistic or Cox regression tests were

employed. Factors with significant correlation identified in the univariate analysis, along with age and sex, were subsequently incorporated in the backward stepwise multivariate regression analysis to identify independent predictive variables. Statistical analysis was performed using the software R 4.3.1 (*R Core Team, 2023*), and all the tests were two-sided, with $P < 0.05$ considered statistically significant. In addition, the R packages used for statistical analyses were as follows: car_3.1-2, carData_3.0-5, tidyr_1.3.1, MASS_7.3–60.2, autoReg_0.3.3, survival_3.6-4, tableone_0.13.2, and haven_2.5.4.

## RESULTS

### ICU-admitted patients with lung cancer who had different 90-day survival demonstrated significant differences in various characteristics

Overall, 269 patients with primary lung cancer diagnosis were admitted to ICUs during the study period. Their median age was 65.0 years (range, 26–85, P25–P75 58.0–71.0), with a significantly higher percentage of males (71.4%). At 90 days after admission, 147 (54.6%) patients survived (Table 1). A comparison between surviving and deceased patients revealed statistically significant differences in various baseline characteristics, including demographics (age and BMI), treatment history, and source of transfer. Furthermore, the characteristics during ICU care, including the disease severity evaluated at admission; ICU diagnoses, including sepsis, respiratory failure, acute respiratory distress syndrome, shock, and AKI; and corresponding treatments all showed significant differences. These results suggest that various characteristics can be associated with different survival outcomes of patients with lung cancer following ICU admission. Overall, patients with 90-day mortality demonstrated more severe conditions before admission, higher number of ICU-related complications, and underwent more intense treatment.

### Predictors of 90-day survival

Cox regression analysis was performed in single- and multivariable settings to further explore potential outcome predictors. In the univariate analysis, most characteristics were correlated with 90-day mortality (Table 2). Subsequently, to identify the independent predictors from the univariate-associated factors, a backward stepwise multivariate regression model approach was performed. The independent predictors identified for 90-day mortality included baseline factors, including age, BMI, chemotherapy history, transfer sources, and whether surgery was performed before admission; complications diagnosed in the ICU; and corresponding treatments, including occurrence of sepsis or respiratory failure, AKI severity, and oxygen support treatment. Generally, these factors were previously correlated with general health conditions, disease severity, and critical conditions during ICU care. Notably, AKI, which was defined and categorized according to the KDIGO criteria (*Kellum & Lameire, 2013*), although only occurred in very small percentage of the patients, was significantly associated with mortality outcomes even in the multivariate analysis. These results highlight the significant role of AKI in the survival prognosis of ICU-admitted patients with lung cancer.

**Table 1  Comparisons of characteristics between ICU lung cancer patients with different 90 days survival outcomes.**

|  | All (n = 269) | Survived at 90 days (n = 147) | 90 days death (n = 122) | P value |
|---|---|---|---|---|
| Demographics |  |  |  |  |
| Age (years), Median [IQR] | 65.0 [58.0, 71.0] | 67.0 [59.5, 71.0] | 63.5 [55.2, 71.0] | 0.043 |
| Gender, n (%) |  |  |  | 0.288 |
| Female | 77 (28.6) | 46 (31.3) | 31 (25.4) |  |
| Male | 192 (71.4) | 101 (68.7) | 91 (74.6) |  |
| BMI (kg/m$^2$), Median [IQR] | 22.2 [20.3, 24.7] | 23.2 [21.2, 25.6] | 21.4 [19.5, 23.7] | <0.001 |
| Treatment history, n (%) |  |  |  |  |
| Target therapy | 55 (20.4) | 17 (11.6) | 38 (31.1) | <0.001 |
| Immunotherapy | 38 (14.1) | 12 (8.2) | 26 (21.3) | 0.002 |
| Chemotherapy | 82 (30.5) | 35 (23.8) | 47 (38.5) | 0.009 |
| Radiotherapy | 19 (7.1) | 5 (3.4) | 14 (11.5) | 0.010 |
| Transferring source, n (%) |  |  |  | <0.001 |
| Operation room | 68 (25.3) | 65 (44.2) | 3 (2.5) |  |
| Emergency department | 16 (5.9) | 4 (2.7) | 12 (9.8) |  |
| Clinical ward | 177 (65.8) | 74 (50.3) | 103 (84.4) |  |
| Other hospitals | 8 (3.0) | 4 (2.7) | 4 (3.3) |  |
| Planned transfer, n (%) | 74 (27.5) | 69 (46.9) | 5 (4.1) | <0.001 |
| Elective or emergency surgery, n (%) |  |  |  | <0.001 |
| No surgery | 161 (59.9) | 56 (38.1) | 105 (86.1) |  |
| Elective | 104 (38.7) | 90 (61.2) | 14 (11.5) |  |
| Emergency | 4 (1.5) | 1 (0.7) | 3 (2.5) |  |
| Severity scores |  |  |  |  |
| SOFA, Median [IQR] | 3.0 [2.0, 6.0] | 3.0 [2.0, 4.0] | 5.0 [3.0, 9.0] | <0.001 |
| APACHE II, Median [IQR] | 13.0 [9.0, 19.0] | 10.0 [8.0, 14.0] | 17.0 [13.0, 25.0] | <0.001 |
| ICU diagnosis, n (%) |  |  |  |  |
| Sepsis | 197 (73.2) | 81 (55.1) | 116 (95.1) | <0.001 |
| ARDS | 56 (20.8) | 14 (9.5) | 42 (34.4) | <0.001 |
| Respiratory failure | 140 (52.0) | 45 (30.6) | 95 (77.9) | <0.001 |
| AKI |  |  |  | <0.001 |
| None | 235 (87.4) | 140 (95.2) | 95 (77.9) |  |
| Grade I | 11 (4.1) | 4 (2.7) | 7 (5.7) |  |
| Grade II | 11 (4.1) | 2 (1.4) | 9 (7.4) |  |
| Grade III | 12 (4.5) | 1 (0.7) | 11 (9.0) |  |
| Shock | 73 (27.1) | 19 (12.9) | 54 (44.3) | <0.001 |
| Delirium | 12 (4.4) | 8 (5.4) | 4 (3.2) | 0.382 |
| Anti-infection treatment, n (%) |  |  |  |  |
| Carbapenems | 92 (34.2) | 26 (17.7) | 66 (54.1) | <0.001 |
| $\beta$-lactam | 128 (47.6) | 58 (39.5) | 70 (57.4) | 0.003 |
| Glycopeptides | 39 (14.5) | 11 (7.5) | 28 (23.0) | <0.001 |
| Tigecycline | 11 (4.1) | 2 (1.4) | 9 (7.4) | 0.026 |
| Echinocandins | 18 (6.7) | 4 (2.7) | 14 (11.5) | 0.004 |
| Triazoles | 49 (18.2) | 12 (8.2) | 37 (30.3) | <0.001 |

**Table 1** (*continued*)

| | All (*n* = 269) | Survived at 90 days (*n* = 147) | 90 days death (*n* = 122) | *P* value |
|---|---|---|---|---|
| Other treatment, *n* (%) | | | | |
| Mechanical ventilation | 125 (46.5) | 40 (27.2) | 85 (69.7) | <0.001 |
| Conventional oxygen therapy | 193 (71.7) | 137 (93.2) | 56 (45.9) | <0.001 |
| Sedation treatment | 97 (36.1) | 26 (17.7) | 71 (58.2) | <0.001 |

**Notes.**
IQR, Interquartile range (P25, P75); BMI, Body Mass Index; SOFA, Sequential Organ Failure Assessment; APACHE, II Acute Physiology And Chronic Health Evaluation II; ARDS, Acute Respiratory Distress Syndrome; AKI, Acute Kidney Injury.

### AKI was a predictor of ICU and in-hospital survival

To further identify the significance of AKI in the prognosis of ICU-admitted patients with lung cancer, multivariate regression analysis was performed using ICU and in-hospital mortality as dependent outcomes. Although the independently correlated factors observed for different outcomes varied, AKI remained independently associated with both survival outcomes (Table 3). These results confirmed that AKI was a significant prognostic factor for the survival of ICU-admitted patients with lung cancer.

### Correlated factors of AKI occurrence in ICU-admitted patients with lung cancer

Owing to the marked significance of AKI for the survival outcomes of ICU-admitted patients with lung cancer, we further explored the correlated factors of AKI. Various characteristics, including transfer source, symptom severity, occurrences of other ICU-related complications, and ICU treatment regimens applied, were different between patients with and without AKI occurrence (Table 4). Owing to the limited sample size, multivariate analysis was not performed. These results indicate that several factors are associated with AKI occurrence, and the major differentiating predictors may be difficult to confirm. However, of note, besides condition severity at admission, ICU treatment choices, particularly anti-infection regimen, may be associated with a higher risk of AKI.

## DISCUSSION

Data obtained from the 37 ICUs of cancer-specialty hospitals revealed that 269 patients with lung cancer were admitted during the 2-month period. The 90-day mortality rate following ICU admission was 45.4%. Patients with 90-day mortality generally exhibited more severe conditions before admission, a higher number of ICU-related complications, and underwent more intense treatment than survivors. Despite its low recorded incidence, AKI was independently associated with ICU, in-hospital, and 90-day mortality outcomes in the multivariate analysis. Moreover, the condition severity at admission and ICU treatment choices, particularly anti-infection regimen, may be associated with a higher risk of AKI.

Previous studies have reported that the in-hospital and short-term mortalities of ICU-admitted patients with solid tumors varied, accounting for mortality rates of 20%–30% (*Martos-Benítez, Soto-García & Gutiérrez-Noyola, 2018*; *Ostermann et al., 2017*; *Puxty et al., 2014*). Although inconsistent results have been reported, it has been suggested that the mortality rate of lung cancer following ICU admission was higher than that of other solid

**Table 2** Regression analysis of predictors for 90 days survival of ICU admitted lung cancer patients.

| Variables | Univariate analysis | | Multivariate analysis | |
|---|---|---|---|---|
| | HR (95% CI) | *P* value | HR (95% CI) | *P* value |
| Demographics | | | | |
| Age (years) | 0.98 (0.96, 1.00) | 0.014 | 0.96 (0.94, 0.98) | <0.001 |
| Gender, *n* (%) | | | | |
| Female | Ref. | | Ref. | |
| Male | 1.18 (0.78, 1.77) | 0.429 | 0.72 (0.47, 1.12) | 0.146 |
| BMI (kg/m$^2$) | 0.90 (0.85, 0.95) | <0.001 | 0.92 (0.87, 0.97) | 0.002 |
| Treatment history | | | | |
| Target therapy | 2.34 (1.59, 3.44) | <0.001 | – | – |
| Immunotherapy | 2.13 (1.38, 3.30) | <0.001 | – | – |
| Chemotherapy | 1.64 (1.14, 2.37) | 0.008 | 0.57 (0.37, 0.87) | 0.009 |
| Radiotherapy | 2.15 (1.23, 3.76) | 0.007 | – | – |
| Transferring source | | | | |
| Operation room | Ref. | | Ref. | |
| Emergency department | 28.38 (8.00, 100.70) | <0.001 | 9.78 (2.22, 43.14) | 0.003 |
| Clinical ward | 19.53 (6.19, 61.62) | <0.001 | 6.41 (1.61, 25.62) | 0.009 |
| Other hospitals | 14.32 (3.20, 64.03) | <0.001 | 2.05 (0.36, 11.81) | 0.423 |
| Planned transfer | 0.08 (0.03, 0.19) | <0.001 | – | – |
| Elective or emergency surgery | | | | |
| No surgery | Ref. | | Ref. | |
| Elective | 0.14 (0.08, 0.25) | <0.001 | 0.38 (0.20, 0.69) | 0.002 |
| Emergency | 1.22 (0.39, 3.84) | 0.738 | 2.28 (0.57, 9.12) | 0.242 |
| Severity scores | | | | |
| SOFA | 1.19 (1.14, 1.24) | <0.001 | 1.05 (1.00, 1.12) | 0.064 |
| APACHE II | 1.07 (1.05, 1.09) | <0.001 | – | – |
| ICU diagnosis | | | | |
| Sepsis | 10.16 (4.47, 23.11) | <0.001 | 2.61 (1.02, 6.67) | 0.045 |
| ARDS | 2.85 (1.96, 4.15) | <0.001 | | |
| Respiratory failure | 4.98 (3.24, 7.66) | <0.001 | 2.16 (1.29, 3.62) | 0.003 |
| AKI | | | | |
| None | Ref. | | | |
| Grade I | 2.03 (0.94, 4.39) | 0.070 | 1.12 (0.50, 2.49) | 0.779 |
| Grade II | 3.82 (1.92, 7.61) | <0.001 | 2.83 (1.32, 6.06) | 0.008 |
| Grade III | 4.93 (2.63, 9.26) | <0.001 | 2.5 (1.23, 5.10) | 0.012 |
| Anti-infection treatment | | | | |
| Carbapenems | 3.09 (2.16, 4.42) | <0.001 | 1.46 (0.97, 2.20) | 0.073 |
| $\beta$-lactam | 1.62 (1.13, 2.32) | 0.009 | – | – |
| Glycopeptides | 2.22 (1.45, 3.40) | <0.001 | – | – |
| Tigecycline | 2.38 (1.21, 4.71) | 0.012 | – | – |
| Echinocandins | 1.96 (1.12, 3.43) | 0.018 | – | – |
| Triazoles | 2.65 (1.80, 3.91) | <0.001 | – | – |

**Table 2** (*continued*)

| Variables | Univariate analysis | | Multivariate analysis | |
|---|---|---|---|---|
| | HR (95% CI) | *P* value | HR (95% CI) | *P* value |
| Other treatment | | | | |
|    Mechanical ventilation | 3.90 (2.64, 5.76) | <0.001 | – | – |
|    Conventional oxygen therapy | 0.15 (0.10, 0.21) | <0.001 | 0.31 (0.20, 0.49) | <0.001 |
|    Sedation treatment | 3.96 (2.75, 5.69) | <0.001 | – | – |

**Notes.**

Cox regression tests were applied to analyze the correlated factors of 90 days death outcomes. Factors with significant correlation identified in univariate analysis, along with age and gender, were then incorporated in backward stepwise multivariate regression analysis to identify independent associated variables.

BMI, Body Mass Index; SOFA, Sequential Organ Failure Assessment; APACHE, II Acute Physiology And Chronic Health Evaluation II; ARDS, Acute Respiratory Distress Syndrome; AKI, Acute Kidney Injury.

**Table 3** Predictors for ICU and in-hospital mortality analyzed using multiple variate logic regression analysis.

| Variables | ICU death | | In-hospital death | |
|---|---|---|---|---|
| | OR (95% CI) | *P*value | OR (95% CI) | *P*value |
| Age (years) | – | – | 0.98 (0.94, 1.03) | 0.465 |
| BMI (kg/m$^2$) | – | – | 1.12 (0.98, 1.29) | 0.090 |
| APACHE II | – | – | 1.05 (0.99, 1.11) | 0.103 |
| Planned transfer | | | 0.04 (0, 0.37) | 0.015 |
| Triazoles | 7.08 (2.57, 21.05) | <0.001 | 5.65 (2.21, 15.22) | <0.001 |
| Target therapy | 3.22 (1.11, 9.63) | 0.032 | 2.56 (0.97, 6.83) | 0.057 |
| Other treatment | | | | |
|    Mechanical ventilation | – | – | 7.96 (1.79, 38.75) | 0.008 |
|    Sedation treatment | 0.49 (0.14, 1.56) | 0.230 | 0.16 (0.04, 0.65) | 0.012 |
|    Conventional oxygen therapy | 0.12 (0.03, 0.40) | 0.001 | 0.12 (0.04, 0.36) | <0.001 |
| ICU diagnosis | | | | |
|    ARDS | 0.32 (0.10, 0.94) | 0.046 | 0.32 (0.10, 0.92) | 0.041 |
|    Respiratory failure | 21.24 (3.24, 432.82) | 0.007 | – | – |
|    AKI | | | | |
|       None | Ref. | | Ref. | |
|       Grade I | 4.24 (0.44, 28.72) | 0.161 | 2.87 (0.48, 14.20) | 0.211 |
|       Grade II | 13.66 (3.08, 68.98) | 0.001 | 16.92 (3.54, 103.46) | 0.001 |
|       Grade III | 9.03 (1.68, 57.26) | 0.013 | 2.73 (0.56, 13.94) | 0.217 |

**Notes.**

Logistic regression tests were applied to analyze the correlated factors of ICU and in-hospital death outcomes. Factors with significant correlation identified in univariate analysis were incorporated in backward stepwise multivariate regression analysis to identify independent associated variables.

BMI, Body Mass Index; APACHE, II Acute Physiology And Chronic Health Evaluation II; ARDS, Acute Respiratory Distress Syndrome; AKI, Acute Kidney Injury.

tumors (*Peng et al., 2021*), reaching approximately 50% (*Andréjak et al., 2011*). Results from this study identified a 45.4% 90-day mortality rate, which is consistent with previous studies. These results confirm that ICU-admitted patients with lung cancer have higher mortality risks.

Various characteristics of admitted patients with lung cancer were identified as factors associated with short-term mortality. However, most of these factors were generally

**Table 4  Comparisons of characteristics and survival outcomes between ICU lung cancer patients with and without AKI.**

|  | All ($n = 269$) | Non-AKI ($n = 235$) | AKI ($n = 34$) | P value |
|---|---|---|---|---|
| Demographics |  |  |  |  |
| Age (year), Median [IQR] | 65.0 [58.0, 71.0] | 65.0 [58.0, 71.0] | 64.5 [57.2, 72.0] | 0.825 |
| Gender, n (%) |  |  |  | 0.482 |
| Female | 77 (28.6) | 69 (29.4) | 8 (23.5) |  |
| Male | 192 (71.4) | 166 (70.6) | 26 (76.5) |  |
| BMI (kg/m$^2$), Median [IQR] | 22.2 [20.3, 24.7] | 22.1 [20.3, 24.6] | 22.8 [20.5, 25.1] | 0.705 |
| Treatment history, n (%) |  |  |  |  |
| Target therapy | 55 (20.4) | 48 (20.4) | 7 (20.6) | 0.983 |
| Immunotherapy | 38 (14.1) | 31 (13.2) | 7 (20.6) | 0.289 |
| Chemotherapy | 82 (30.5) | 70 (29.8) | 12 (35.3) | 0.514 |
| Radiotherapy | 19 (7.1) | 15 (6.4) | 4 (11.8) | 0.276 |
| Transferring source, n (%) |  |  |  | 0.005 |
| Operation room | 68 (25.3) | 66 (28.1) | 2 (5.9) |  |
| Emergency department | 16 (5.9) | 11 (4.7) | 5 (14.7) |  |
| Clinical ward | 177 (65.8) | 151 (64.3) | 26 (76.5) |  |
| Other hospitals | 8 (3.0) | 7 (3.0) | 1 (2.9) |  |
| Planned transfer, n (%) | 74 (27.5) | 72 (30.6) | 2 (5.9) | 0.003 |
| Elective or emergency surgery, n (%) |  |  |  | 0.003 |
| No surgery | 161 (59.9) | 133 (56.6) | 28 (82.4) |  |
| Elective | 104 (38.7) | 99 (42.1) | 5 (14.7) |  |
| Emergency | 4 (1.5) | 3 (1.3) | 1 (2.9) |  |
| Severity scores |  |  |  |  |
| SOFA, Median [IQR] | 3.0 [2.0, 6.0] | 3.0 [2.0, 6.0] | 8.0 [4.2, 11.8] | <0.001 |
| APACHE II, Median [IQR] | 13.0 [9.0, 19.0] | 12.0 [9.0, 17.5] | 20.5 [14.2, 26.0] | <0.001 |
| ICU diagnosis |  |  |  |  |
| Sepsis | 197 (73.2) | 165 (70.2) | 32 (94.1) | 0.003 |
| ARDS | 56 (20.8) | 42 (17.9) | 14 (41.2) | 0.002 |
| Respiratory failure | 140 (52.0) | 113 (48.1) | 27 (79.4) | 0.001 |
| Shock | 73 (27.1) | 48 (20.4) | 25 (73.5) | <0.001 |
| Delirium | 12 (4.5) | 11 (4.6) | 1 (2.9) | 0.988 |
| Anti-infection treatment, n (%) |  |  |  |  |
| Carbapenems | 92 (34.2) | 71 (30.2) | 21 (61.8) | <0.001 |
| $\beta$-lactam | 128 (47.6) | 112 (47.7) | 16 (47.1) | 0.948 |
| Glycopeptides, | 39 (14.5) | 31 (13.2) | 8 (23.5) | 0.120 |
| Tigecycline | 11 (4.1) | 7 (3.0) | 4 (11.8) | 0.037 |
| Echinocandins | 18 (6.7) | 17 (7.2) | 1 (2.9) | 0.710 |
| Triazoles | 49 (18.2) | 37 (15.7) | 12 (35.3) | 0.006 |
| Other treatment, n (%) |  |  |  |  |
| Mechanical ventilation | 125 (46.5) | 100 (42.6) | 25 (73.5) | 0.001 |
| Conventional oxygen therapy | 193 (71.7) | 176 (74.9) | 17 (50.0) | 0.003 |
| Sedation treatment | 97 (36.1) | 77 (32.8) | 20 (58.8) | 0.003 |

**Table 4** (*continued*)

|  | All (*n* = 269) | Non-AKI (*n* = 235) | AKI (*n* = 34) | *P* value |
|---|---|---|---|---|
| Survival outcomes |  |  |  |  |
| ICU mortality, *n* (%) | 32 (11.9) | 18 (7.7) | 14 (41.2) | <0.001 |
| In-hospital mortality, *n* (%) | 41 (15.2) | 25 (10.6) | 16 (47.1) | <0.001 |
| 90 days mortality, *n* (%) | 122 (45.4) | 95 (40.4) | 27 (79.4) | <0.001 |

**Notes.**
IQR, Interquartile range (P25, P75); BMI, Body Mass Index; SOFA, Sequential Organ Failure Assessment; APACHE, II Acute Physiology And Chronic Health Evaluation II; ARDS, Acute Respiratory Distress Syndrome.

correlated with disease severity evaluation, and a clear pattern of predictors was difficult to recognize. Interestingly, in this study, increased BMI was suggested to be associated with better survival prognosis. This finding is not surprising, as higher BMI is associated with better nutrition and health status. In a previous study, a BMI cutoff of 25 kg/m$^2$ was suggested for the ICU admission evaluation of patients with cancer (*Zarogoulidis et al., 2013*). However, the average BMI of the study population was relatively higher than 24 kg/m$^2$; therefore, increased BMI could be associated with a better prognosis. In this study, transfer sources were found to be associated with survival outcomes. Patients with cancer are typically admitted to the ICU due to oncology, treatment-related complications, or precautious postoperative care (*Biskup et al., 2017*). The source of transfer may be correlated with the severity of complications, indicating different prognostic risks. The choice of anti-infection regimen was correlated with survival outcomes. This finding can be attributed to two reasons. One finding is that the more intense treatment indicates more severe clinical conditions. Additionally, anti-infection regimens were primarily empirical, and certain medications may increase the risks of organ injuries. Previous studies have suggested an association between early empirical antifungal therapy and survival rates in patients with cancer (*Kanj et al., 2022*).

The results of this study indicate that AKI is significant in predicting adverse survival outcomes. AKI is a frequent complication of critical illness and in patients with cancer and is associated with worse survival outcomes (*Córdova-Sánchez et al., 2021*; *Hoste et al., 2015*; *Kang et al., 2019*; *Seylanova et al., 2020*). It has been suggested to be a key factor in risk stratification and predicting mortality in ICU patients (*Huang et al., 2021*; *Libório et al., 2011*; *Wan et al., 2020*; *Wiersema et al., 2019*). Previous studies have suggested that AKI timing onset and progression help to predict the outcomes of sepsis and in-hospital mortality (*Wang et al., 2023*). Recognition and prompt notification of AKI may help improve the outcomes of patients with critical illness (*Atia et al., 2023*). From a mechanistic perspective, AKI increased the risks of other adverse outcomes, including stroke, cardiovascular disease, and upper gastrointestinal hemorrhage (*Fortrie, de Geus & Betjes, 2019*). These results further suggested that AKI evaluation and prevention could serve as a critical strategy in the ICU admission and management of patients with lung cancer.

This study had some limitations. First, this study was conducted in a relatively short period and only in the ICUs of cancer-specialty hospitals, which may have caused the selection bias of patients. Moreover, owing to the nature of an observational study, the data

quality should be further improved. For example, the incidence of ICU-associated AKI in this study was significantly lower than previously reported, and the data of corresponding treatment for AKI were missing. The small number of AKI events recorded may limit the certainty of the conclusion, and the small number of AKI events limited the analysis for the potential correlator analysis. Conversely, the statistically independent influence of AKI on mortality being detected even in this small sample size of events may further emphasize that AKI could be among the most critical predictors of prognosis in ICU-admitted patients with lung cancer. Furthermore, as the data were extracted from observational study records from the ICU, certain known prognostic factors, including staging and histological typing of cancer, and performance score were missing. These may introduce further uncertainty to the conclusion. Nevertheless, this study collected data with a relatively large sample size from multiple centers in China. These findings suggest that more large-scale population studies are warranted to clarify the predictors of survival outcomes following ICU care of patients with lung cancer.

## CONCLUSIONS

The results of this study suggest that predictive tools are essential for ICU-admitted patients with lung cancer owing to their high mortality risks. AKI prediction and prevention should be prioritized in these patients.

### Funding
The authors received no funding for this work.

### Competing Interests
The authors declare there are no competing interests.

### Author Contributions
- Jue Shen conceived and designed the experiments, prepared figures and/or tables, authored or reviewed drafts of the article, and approved the final draft.
- Changsong Wang performed the experiments, prepared figures and/or tables, and approved the final draft.
- Gang Ma performed the experiments, prepared figures and/or tables, and approved the final draft.
- Hong-Zhi Wang performed the experiments, prepared figures and/or tables, and approved the final draft.
- Xuezhong Xing analyzed the data, prepared figures and/or tables, and approved the final draft.
- Biao Zhu analyzed the data, prepared figures and/or tables, and approved the final draft.
- Jianghong Zhao analyzed the data, prepared figures and/or tables, and approved the final draft.

- Donghao Wang conceived and designed the experiments, prepared figures and/or tables, authored or reviewed drafts of the article, and approved the final draft.
- Mingou Cui conceived and designed the experiments, prepared figures and/or tables, authored or reviewed drafts of the article, and approved the final draft.

## Human Ethics

The following information was supplied relating to ethical approvals (*i.e.*, approving body and any reference numbers):

This study was approved by Ethic committee of Tianjin Medical University Cancer Institute and Hospital (bc2021065).

## Data Availability

Raw data is available in the Supplemental Files.

## Supplemental Information

Supplemental information for this article can be found online at http://dx.doi.org/10.7717/peerj.19885#supplemental-information.

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
