# Peer review of "Survival predictors of lung cancer patients in ICU: the importance of acute kidney injury prediction and prevention"

_PeerJ, doi:10.7717/peerj.19885_

## Round 0.1 · original submission · Major Revisions

· Academic Editor

Major Revisions

All 3 reviewers have provided suggestions for major revisions. Please address all of their concerns.

Reviewer 1 ·

Basic reporting

- Spell out all abbreviations at first use (e.g., ICU, AKI) for clarity.
- A thorough proofreading could help address any minor grammatical issues.
- Consider enhancing the figures and tables with more detailed legends to improve readability.

Experimental design

non

Validity of the findings

- If data on performance status and tumor stage were not collected, it may be worth noting this as a limitation, as these are known prognostic factors. The authors may wish to include data on delirium of ICU stay in the results, as these factors can impact outcomes.
- The criteria for variable selection in the multivariate analysis should be more explicitly stated.
- The importance of AKI is emphasized, but the response to AKI (e.g. choice of dialysis) may differ depending on the stage of tumor. This point also needs to be considered. Furthermore, the analysis is univariate and based on a small number of AKI subjects, which may lead to weak conclusions.
- Rather than concluding that " predictors of ICU lung cancer patients are complicated," aim to provide more specific insights on identified predictors and their clinical significance.

Additional comments

non

Reviewer 2 ·

Basic reporting

Language & Professionalism: The manuscript is written in clear and professional English. The terminology is appropriate for the audience, and the technical content is communicated unambiguously.

Literature references: The introduction provides an adequate background but lacks a more thorough discussion of previous studies on lung cancer patient outcomes in ICU, especially in the context of AKI. The literature should be expanded to better highlight the knowledge gap this study aims to fill.

Structure: The manuscript follows a standard scientific structure. The figures and tables are well-labeled and relevant to the study.

Experimental design

Scope & Originality: The study fits within the aims and scope of the journal. It addresses a critical gap in the literature by focusing on survival predictors, particularly AKI, among lung cancer patients admitted to the ICU. The research question is clearly defined and relevant: identifying predictors for ICU survival, with an emphasis on the role of AKI.

Data Collection and Variables: The study includes a wide range of variables, but the methods section lacks sufficient detail on how AKI was diagnosed across centers. Were there standardized diagnostic criteria, or did the criteria vary between sites? This is critical for interpreting the results on AKI incidence and tis correlation with mortality.

Statistical Methods: The statistical methods used are generally appropriate. However, the authors mention significant multicollinearity between some variables. It would be helpful to explain how this issue was addressed, as multicollinearity could affect the robustness of the regression models.

Validity of the findings

Conclusion Alignment: The emphasis on AKI as a predictor of mortality is compelling, but the study suffers from small sample sizes in some subgroups (e.g., patients with AKI), which could reduce the generalizability of the findings. The limitations section mentions this briefly, but a more detailed discussion of how sample size impacts the study’s conclusions would be beneficial.

Additional comments

General Comments:

Strengths: The study is commendable for its large multi-center design, which adds robustness to the findings. The focus on AKI as a critical factor in ICU outcomes for lung cancer patients is novel and relevant to current clinical challenges.

Weaknesses: The study would benefit from a more standardized approach to diagnosing and categorizing AKI across centers. Additionally, the relatively short data collection period (two months) and the limited number of patients with AKI make it difficult to draw firm conclusion.

Reviewer 3 ·

Basic reporting

no comment

Experimental design

no comment

Validity of the findings

no comment

Additional comments

The authors conducted a clinical study to investigate the variables associated with short-term mortality in ICU patients with lung cancer. They found that several factors, including baseline clinical characteristics, comorbidities, and treatment modalities, are associated with a worse prognosis in this patient population. Among these factors, they identified AKI as an important prognostic factor that should be managed as a priority. Some issues need to be addressed before this manuscript can be accepted.

1. The English language in the manuscript needs improvement. Several paragraphs were written in a Chinese-English style and should be edited by a proficient native English speaker.
2. The primary endpoint of this study is 90-day mortality; therefore, I believe it is a longitudinal study, not a cross-sectional study.
3. Why did you include individuals older than 14 years? This may result in a heterogeneous population that includes both pediatric and adult patients.
4. You should only state the study findings in the results section. Statements such as “Predictors of ICU lung cancer patients are complicated. A more precise and instructive model needs to be developed. Prediction and prevention of AKI should be prioritized for lung cancer patients admitted to the ICU” should not be included in the results section.
5. There is no conclusion section in your abstract.
6. I cannot understand the meaning of this sentence. Why has the progression of cancer treatment and ICU management significantly increased the number of patients requiring ICU care?
7. Some abbreviations, such as ARF, SOFA, and AKI, should be spelled out in full when they first appear.
8. The threshold of p-value should be mentioned in the statistical section.
9. The definition of acute kidney injury should be clearly described.
10. The categories of lung cancer should be presented. For example, small cell lung cancer and non-small cell lung cancer.
11. You should describe how the multivariate models were constructed in this study.

---

## Round 0.2 · Minor Revisions

· Academic Editor

Minor Revisions

In addition to the comments by the authors, please deposit your code and raw data in Zenodo or another appropriate database with a DOI. Additionally, statistical packages used (including R packages) should be cited with the appropriate versions and references.

Reviewer 1 ·

Basic reporting

・Abbreviations: As previously noted, key abbreviations such as “ICU (Intensive Care Unit)” and “AKI (Acute Kidney Injury)” are now spelled out upon first use, which improves clarity for the reader.

・Language and Proofreading: The overall language has been refined, and the manuscript now shows evidence of thorough proofreading. Minor grammatical issues and ambiguous expressions have been resolved. In addition, the figure and table legends have been enhanced, making the data presentation much clearer.

Experimental design

non

Validity of the findings

・Acknowledgment of Missing Prognostic Factors: The authors now clearly acknowledge that certain known prognostic factors (e.g., performance status, tumor stage, and histological typing) were not collected. This limitation is discussed transparently in the manuscript.

・Delirium Data: Data on delirium during ICU stay have been included in the results, addressing previous suggestions to consider this factor.

・AKI Analysis: Although the incidence of AKI remains low, the manuscript emphasizes its significance as an independent predictor of outcome. The authors also discuss the limitation regarding missing detailed treatment data (such as dialysis choices) and acknowledge the constraints posed by the small number of AKI events.

・Specificity in Conclusions: Rather than a broad statement that “predictors of ICU lung cancer patients are complicated,” the conclusion now specifically focuses on the need to prioritize the prediction and prevention of AKI in this patient population, providing clearer clinical insights.

Additional comments

non

Reviewer 2 ·

Basic reporting

The authors have adequately addressed my concerns and have updated the manuscript accordingly.

Experimental design

The authors have adequately addressed my concerns and have updated the manuscript accordingly.

Validity of the findings

The authors have adequately addressed my concerns and have updated the manuscript accordingly.

Reviewer 3 ·

Basic reporting

The authors have addressed the mentioned issues, and I consider this manuscript acceptable for publication.

Experimental design

N/A

Validity of the findings

N/A

Additional comments

N/A

---

## Round 0.3 · Minor Revisions

· Academic Editor

Minor Revisions

The authors must cite the R packages they used in the manuscript.

The R files include comments in Chinese that must be translated for an English-speaking audience.

---

## Round 0.4 · Minor Revisions

· Academic Editor

Minor Revisions

Thank you for updating the comments to english. I still do not see that the R packages used in the code have appropriate versions in either the code or the manuscript. Will you please update it or point me to the lines in the manuscript and the code where that is the case? Thank you.

---

## Round 0.5 · accepted · Accept

· Academic Editor

Accept

Thank you for adding the R package version numbers. This manuscript is now ready for acceptance.